# Vision Measurement Method Based on Plate Glass Window Refraction Model in Tunnel Construction

**DOI:** 10.3390/s24010066

**Published:** 2023-12-22

**Authors:** Zhen Wu, Junzhou Huo, Haidong Zhang, Fan Yang, Shangqi Chen, Zhihao Feng

**Affiliations:** 1School of Mechanical Engineering, Dalian University of Technology, Dalian 116024, China; wuzhen0808@mail.dlut.edu.cn (Z.W.); zhanghaidong@mail.dlut.edu.cn (H.Z.); youngfine@mail.dlut.edu.cn (F.Y.); 18004087013@163.com (S.C.); fengzhihao95@163.com (Z.F.); 2State Key Laboratory of High-Performance Precision Manufacturing, Dalian 116024, China

**Keywords:** tunnel construction, visual measurement, plate glass refraction, distortion correction

## Abstract

Due to the harsh environment of high humidity and dust in tunnel construction, the vision measurement system needs to be equipped with an explosion-proof glass protective cover. The refractive effect of the plate glass window invalidates the pinhole model. This paper proposes a comprehensive solution for addressing the issue of plane refraction. First, the imaging model for non-parallel plane refraction is established based on dynamic virtual focal length and the Rodriguez formula. Further, due to the failure of the epipolar constraint principle in binocular vision systems caused by plane refraction, this paper proposes the epipolar constraint model for independent refractive plane imaging. Finally, an independent refraction plane triangulation model is proposed to address the issue of triangulation failure caused by plane refraction. The RMSE of the depth of field errors in the independent refraction plane triangulation model is 2.9902 mm before correction and 0.3187 mm after correction. The RMSE of the positioning errors before and after correction are 3.5661 mm and 0.3465 mm, respectively.

## 1. Introduction

A Tunnel Boring Machine (TBM) is an extensive piece of equipment utilized in tunnel construction. It combines various functions including excavation, removal of debris, and lining. TBMs are widely used in many fields such as railroads, highways, water diversion projects, and municipal pipelines. In the TBM digging process, the cutter is subjected to complex and variable loads. The cutter consumes a lot during the construction process and requires frequent replacement [1,2]. The tool change area [3] is shown in Figure 1. The cutter change area is very narrow and not conducive to manual work. At present, TBM tool-changing work is dangerous and inefficient [4]. This has led to an urgent demand for robotic tool-changing technology in the field of tunnel construction. Currently, manual operations are primarily relied upon for tool detection and replacement due to limitations in technological development. During the construction process of TBMs, there are considerable safety hazards related to manual operations in harsh construction environments. These environments include deep burial, high water pressure, and high levels of dust, which can potentially result in serious safety accidents such as casualties. Thus, achieving intelligent tool-changing in harsh and complex tunnel construction environments requires precise positioning and accurate measurement of the tool system.

Machine vision measurement technology has advantages over other measurement technologies such as non-contact measurement, remote manipulation, information-rich, fast response, and accuracy [5,6,7]. The environment of the TBM tool change bin is often characterized by high pressure [8], high humidity, and dust. To ensure safety, the camera inside the tool changing bin needs to be equipped with an explosion-proof glass protective cover [9,10]. However, the traditional pinhole camera model cannot eliminate the refractive distortion caused by glass. The existing distortion models with a single factor (only considering the lens) cannot describe the imaging patterns of cameras under planar refraction [11].

This paper presents a comprehensive solution for binocular measurement under the refraction of plate glass. The specific contributions are as follows:(1)The non-parallel plane refractive camera imaging model is established based on dynamic virtual focal length and the Rodriguez formula.(2)The main problems are the epipolar constraint failure and triangulation failure under refraction images. This paper proposes an epipolar constraint model and triangulation method based on independent refraction planes.

The remainder of this paper is structured as follows. Section 2 introduces the related work. Section 3 describes, in detail, the model for image distortion correction under non-parallel plane refraction, the model for imaging epipolar constraint on independent refractive surfaces, and the triangulation method for independent refraction planes. Section 4 validates the effectiveness of the proposed method by conducting a series of physical experiments. Section 5 discusses the relevant issues. Section 6 presents the conclusion.

## 2. Related Work

The refraction effect of glass severely affects the accuracy of visual measurements. Scholars have conducted in-depth research on the characterization and elimination of refractive distortion. In the calibration of visual measurement system parameters under planar refraction, Treibitz [12] studied the trajectory of camera viewpoints under planar refraction environments. In addition, Treibitz proposed an imaging model for a camera based on dynamic virtual focal length. The measurement accuracy of this model is significantly greater compared to single-viewpoint models. However, this model requires prior knowledge of the depth information of the target object. The camera’s optical axis must also be parallel to the normal vector of the refractive surface. Agrawal [13] established an imaging model for multiplane refraction. The multi-plane refraction imaging model has been proven to have strong similarities to the single-plane refraction imaging model.

Chen [14] introduced a calibration method for determining the normal vector of the refractive surface as well as the thickness of the glass. The experiment demonstrates that this method exhibits high accuracy and robustness. Shimizu [15,16] proposed a camera depth estimation method that utilizes the parallel plane refraction effect. However, this method requires a complex calibration setup. Gong [17] proposed a 3D reconstruction model based on plane refraction correction. This method applies to the scene of any number and direction of plate glass. He also proposed a method for flexibly determining the normal vector of the refractive surface without any auxiliary devices. Ke [18] examined how the thickness of a glass plate affects the resulting image. He used a target extremely close to the plate glass to obtain the normal vector of the refractive surface and the distance between the plate glass and the camera’s optical center. Huang S [19] proposed a multi-camera calibration method based on planar refraction. Multiple cameras can capture calibration images from different angles. This calibration method has high accuracy. The basic geometric relationships based on plane refraction can be used to eliminate refraction errors. Shortis et al. [20] employ corrective lenses or dome glass to eliminate refractive distortion, guaranteeing that the light rays from every point are perpendicular to the refractive surface. This situation can still use a single viewpoint model. However, this method requires extremely high accuracy. In practical applications, manufacturing errors and installation deviations of lenses often lead to other refractive errors.

Additionally, scholars have conducted research on stereo matching in binocular vision under plane refraction. Yamashita [21] considered the refraction effect and proposed a model for underwater 3D reconstruction. However, he did not consider the issue of polar constraint failure under refractive effects. Huang et al. [22] proposed a multi-plane refractive imaging model. He proposed the theory of dynamic epipolar constraint, which solved the problem that the epipolar of stereo matching under plane refraction is a curve. This method verifies the accuracy of matching points. However, the camera’s optical axis remains parallel to the normal vector of the refractive surface. Gedge [23] studied the effect of planar refraction on underwater imaging and stereo matching. He proposed a 3D reconstruction model when the camera’s optical axis is not parallel to the normal vector of the refractive surface, and the curve equations for the epipolar constraint in stereo matching are calculated.

In summary, a situation of approximate correction in the research of visual measurement methods under plane refraction remains. Many scholars have built plane refraction imaging models that are too complex, with too many introduced external parameters. The model is not concise and requires large computational efforts.

This paper investigates the problem of error correction for vision measurement systems considering plane refraction and presents a complete approach to visual measurements. This method includes image distortion correction, refraction plane normal vector solving, polar constraint modeling of refraction images, triangulation modeling of refraction imaging, and refraction error measurement.

## 3. Methodology

The traditional pinhole camera model based on perspective imaging leads to vision measurement errors due to refraction distortion. A new refractive imaging model needs to be established to eliminate refractive errors. Ideally, the camera’s optical axis is parallel to the normal vector of the refractive plane. Due to installation and manufacturing errors, the camera’s optical axis is not guaranteed to be parallel to the normal vector of the refraction plane. An image distortion correction model is established based on the concept of dynamic virtual focal length in a parallel plane. It complements the non-parallel plane camera rotation model to achieve correction. The main flow of this paper is shown in Figure 2.

### 3.1. Two-Dimensional Measurement with Monocular Vision under Non-Parallel Plane Refraction

#### 3.1.1. Finding the Normal Vector of a Refraction Plane

In Figure 3, the camera’s optical axis is not parallel to the normal vector ***n*** of the refractive plane. Based on the expansion offset effect of plane refraction, the normal vector of the refraction plane of the plate glass is determined.

The camera coordinate system is established with *O* as the origin. The *M_g_*_1_(*u_g_*_1_, *v_g_*_1_) and *M_g_*_2_(*u_g_*_2_, *v_g_*_2_) points are the refracted imaging points. *M*_1_(*u*_1_, *v*_1_) and *M*_2_(*u*_2_, *v*_2_) are the image points without refraction. *M_c_*(*u_c_*, *v_c_*) is the distortion center point. One can maintain the calibration board posture and take two separate images, one directly imaged and the other imaged through glass. These two images can then be used to get the refraction plane normal vector. As depicted in Figure 3a, the vector *M_c_O* is parallel to the normal vector n of the refraction plane in the camera coordinate system. As depicted in Figure 3b, the intersection of two straight lines in the ideal state is the distortion center point. The intersection point between the normal vector of the glass refractive plane passing through the optical center and the image plane is the distortion center. Therefore, different glass attitudes can cause significant differences in the location of distortion centers. In practical implementation, it is necessary to calculate multiple lines to ensure the accuracy of the distortion center. The point closest to all the lines is taken as the distortion center.

The equation of the line connecting the refracted image point and the non-refracted image point is:(1)$$x(v_{gi}-v_{i})+y(u_{i}-u_{gi})+v_{i}u_{gi}-u_{i}v_{gi}=0$$

The distortion center *M_c_*(*u_c_*, *v_c_*) is the point closest to all straight lines, which can be solved by linear least square method [23]. The objective function is established as follows:(2)$$\arg\underset{\left(u_{c},v_{c}\right)}{\min}{\sum_{i=1}^{K}\left(\frac{u_{c}(v_{gi}-v_{i})+v_{c}(u_{i}-u_{gi})+v_{i}u_{gi}-u_{i}v_{gi}}{{\left\|\left(v_{gi}-v_{i}),(u_{i}-u_{gi}\right)\right\|}_{2}}\right)}^{2}$$

The normal vector of the refraction plane of the plate glass is
(3)$$n=[\frac{u_{c}-u_{0}}{fdx},\frac{v_{c}-v_{0}}{fdy},1]$$

#### 3.1.2. Modeling of Camera Rotation under Non-Parallel Plane Refraction

The image distortion correction model under parallel plane refraction is established based on the dynamic virtual focal length. After determining the refraction plane normal vector ***n***, the position of the inclined plate glass in the camera coordinate system can be accurately described. It can establish a camera rotation model under non-parallel plane refraction. In Figure 4, the virtual camera coordinate system (blue dotted line in the figure) is established with point *O* as the origin. The tilt of plate glass can be viewed as a rotation centered on the rotation axis past the origin *O*.

The unit vector ***v*** of the rotation axis can be found from the camera optical axis direction vector ***w*** and the refraction plane normal vector ***n*** as follows:(4)$$v=\frac{w\times n}{\left|w\times n\right|}$$

The angle of rotation is as follows:(5)$$\cos\lambda =\frac{w\cdot n}{\left|w\right|\cdot \left|n\right|}$$

The virtual camera coordinate system can be obtained by rotating the real camera coordinate system around the rotation axis ***v*** using ***λ*** angle. First, the *Q_z_*(*x_z_*, *y_z_*, *z_z_*) of the camera coordinate system is transformed into the virtual camera coordinate system. It follows from the Rodriguez equation:(6)$$q_{z}^{\prime }=\sin\lambda (v\times q_{z})+(1-\cos\lambda )(v\cdot q_{z})v+q_{z}\cos\lambda$$
where *q_z_* = [*x_z_*, *y_z_*, *z_z_*]^T^, *q_z_*′ = [*q_z_*_1_, *q_z_*_2_, *q_z_*_3_]. The *q_z_*′ is the coordinate of *Q_z_* in the virtual camera coordinate system. In the virtual camera coordinate system, the intersection of the line *OQ_z_* with the virtual image plane is the image point *Q*_2_(*x*_2_, *y*_2_, *z*_2_).
(7)$$\left\{\begin{array}{c} \frac{x_{2}}{q_{z1}}=\frac{y_{2}}{q_{z2}}=\frac{z_{2}}{q_{z3}}\\ z_{2}=-f \end{array}\right.$$

The following is from Equation (7):(8)$$\left\{\begin{array}{c} x_{2}=-\frac{f\cdot q_{z1}}{q_{z3}}\\ y_{2}=-\frac{f\cdot q_{z2}}{q_{z3}} \end{array}\right.$$

In conclusion, the coordinates of the image point in the virtual camera coordinate system can be obtained. The correction for non-parallelism between the camera optical axis and the normal vector of the refracting surface can be realized.

#### 3.1.3. Modeling of Camera Imaging under Non-Parallel Plane Refraction

In Figure 5, the intersection point between the camera optical axis and the measurement plane in the camera coordinate system is *Q*_0_. *Q*_1_ is any point on the measurement plane. The image point of *Q*_1_ refracted by the plane on the virtual imaging plane is *Q*_2_. The main point of the virtual camera is *Q*_4_. The incident angle is *θ_a_*_1_. The distance between line *OQ*_2_ and line *L* is denoted as *d_w_*_1_. In the camera coordinate system, the coordinates of *Q*_0_ are (0, 0, *h*). In the virtual camera coordinate system, the coordinates of *Q*_0_, *Q*_1_, *Q*_2_, *Q*_3_, and *Q*_4_ are *Q*_0_(*x*_0_, *y*_0_, *z*_0_), *Q*_1_(*x*_1_, *y*_1_, *z*_1_), *Q*_2_(*x*_2_, *y*_2_, *z*_2_), *Q*_3_(*x*_3_, *y*_3_, *z*_3_), and *Q*_4_(*x*_4_, *y*_4_, *z*_4_). The coordinates of *Q*_0_ and *Q*_2_ in the virtual camera coordinate system can be determined based on the camera rotation model. The coordinates of *Q*_4_ in the virtual camera coordinate system are (0, 0, −*ƒ*). In the camera coordinate system, the normal vector of the refractive surface is ***n*** = [n_1_, n_2_, n_3_]. The normal vector of the measurement plane is ***w*** = [0, 0, 1]. The ***w*** can be corrected by the camera rotation model to obtain the normal vector of the measurement plane in the virtual camera coordinate system as ***w*_1_** = [−*n*_1_, −*n*_2_, *n*_3_]. 

In the virtual camera coordinate system, the normal vector ***w*_1_** of the measurement plane and a certain point *Q*_0_(*x*_0_, *y*_0_, *z*_0_) in the measurement plane are known. From any point *Q*(*x*, *y*, *z*) in the measurement plane:(9)$$\vec{QQ_{0}}\cdot w_{1}=0$$

The point *Q* coordinate satisfies the following:(10)$$a(x-x_{0})+b(y-y_{0})+c(z-z_{0})=0$$

After this, the equation of the measurement plane can be obtained as follows:(11)ax+by+cz+d=0
where *a* = −*n*_1_, *b* = −*n*_2_, *c* = *n*_3_, *d* = −(*ax*_0_ + *by*_0_ + *cz*_0_).

In the virtual camera coordinate system, the distances *Q*_2_*Q*_4_ and *Q*_3_*Q*_4_ on the virtual imaging surface are *r*_2_ and *r*_3_, respectively.

This can be obtained from the geometric relationship in Figure 5:(12)$$\left\{\begin{array}{c} \tan\theta_{a1}=\frac{r_{2}}{f}\\ d_{w1}=d\sin\theta_{a1}(1-\frac{\sqrt{1-\sin^{2}\theta_{a1}}}{\sqrt{n_{p}^{2}-\sin^{2}\theta_{a1}}})\\ r_{3}-r_{2}=\frac{d_{w1}}{\cos\theta_{a1}} \end{array}\right.$$

Since the line *Q*_2_*Q*_4_ is collinear with the line *Q*_3_*Q*_4_, it follows:(13)$$\frac{x_{2}}{x_{3}}=\frac{y_{2}}{y_{3}}$$

By combining Equations (12) and (13), *x*_3_ and *y*_3_ can be obtained. The coordinates of *Q*_3_ can be obtained.

In the virtual camera coordinate system, the line *OQ*_2_ is parallel to the line *L*. In Figure 5, the equation for line *L* is as follows:(14)$$\frac{x-x_{3}}{x_{2}}=\frac{y-y_{3}}{y_{2}}=\frac{z-z_{3}}{z_{2}}$$

The intersection point between the line *L* and the measurement plane is *Q*_1_. The coordinates of *Q*_1_ in the virtual camera coordinate system can be obtained by combining Equations (11) and (14). Its converted coordinate in the camera coordinate system is as follows:(15)$$q_{1}=-\sin\lambda (v\times q_{1}^{\prime })+(1-\cos\lambda )(v\cdot q_{1}^{\prime })v+q_{1}^{\prime }\cos\lambda$$
Here, ***q*_1_′** = [*x*_1_, *y*_1_, *z*_1_]^T^, *q*_1_ = [*q*_11_, *q*_12_, *q*_13_], (*q*_11_, *q*_12_, *q*_13_) are the coordinates of *Q*_1_ in the camera coordinate system.

### 3.2. Epipolar Constraint Modeling of Binocular Vision System under Nonparallel Plane Refraction

Based on the geometric relationship of plane refraction, the epipolar constraint model of the binocular vision system under non-parallel plane refraction is derived. The virtual camera coordinate system is introduced, and the conversion relationship between the camera coordinate system and the virtual camera coordinate system is shown in Figure 5. The coordinate system of the left camera is denoted as *O_l_X_l_Y_l_Z_l_*. The coordinate system of the left virtual camera is denoted as *O_l_X_cl_Y_cl_Z_cl_*. The coordinate system for the right camera is denoted as *O_r_X_r_Y_r_Z_r_*. The coordinate system for the right virtual camera is denoted as *O_r_X_cr_Y_cr_Z_cr_*. The optical centers of the two cameras are *O_l_* and *O_r_*. The base distance between the two optical centers is *a*. The focal length of the cameras is *ƒ*. The left and right virtual imaging planes are maintained parallel to the plate glass.

In Figure 6b, the two camera refraction planes are independent as shown. Based on the virtual camera coordinate system, the epipolar constraint model of independent refraction plane imaging is established. The distortion centers of the left and right virtual imaging planes are denoted as *F_cl_* and *F_cr_*, respectively. The thickness of plate glass is *d*. *O_l_* is the origin of the left virtual camera coordinate system *O_l_X_cl_Y_cl_Z_cl_*. The left virtual camera coordinate system serves as the world coordinate system. *F*_1_(*x*_1_, *y*_1_, *z*_1_) is any point on the left virtual imaging plane. The line *O*_l_*F*_1_ intersects the front of the glass at point *F*_2_, and after being refracted by the left plate glass, intersects the back of the glass at point *F*_3_. After this, take any point *F_h_* (*x_h_*, *y_h_*, *z_h_*) on the refraction ray and set *z_h_* = *h*. The inverse extension of the line *F*_3_*F_h_* intersects the left virtual imaging plane at point *F*_4_. Set the refraction planes in the left and right cameras as **Π_1_**, **Π_2_**. As a result, the intersection of the refraction plane **Π_1_** with the left virtual imaging plane is the line *F*_1_*F*_4_. The distortion center *F_cl_* (*x_cl_*, *y_cl_*, *z_cl_*) of the left virtual imaging plane is on line *F*_1_*F*_4_. The coordinates of *F*_4_ is (*x*_4_,*y*_4_,*z*_4_). The incident angle is *θ_a_*_2_. The lengths of lines *F_cl_F*_1_ and *F_cl_F*_4_ are *R*_1_ and *R*_2_, respectively. The distance between the incident and refraction rays is *d_w_*_2_.

From the geometric relationship in Figure 6:(16)$$\left\{\begin{array}{c} R_{1}^{2}={\left(x_{1}-x_{cl}\right)}^{2}+{\left(y_{1}-y_{cl}\right)}^{2}\\ R_{2}^{2}={\left(x_{4}-x_{cl}\right)}^{2}+{\left(y_{4}-y_{cl}\right)}^{2}\\ \tan\theta_{a2}=\frac{R_{1}}{f}\\ d_{w2}=d\sin\theta_{a2}(1-\frac{\sqrt{1-\sin^{2}\theta_{a2}}}{\sqrt{n_{p}^{2}-\sin^{2}\theta_{a2}}})\\ R_{1}-R_{2}=\frac{d_{w2}}{\cos\theta_{a2}}\\ \frac{x_{1}}{x_{4}}=\frac{y_{1}}{y_{4}}=\frac{z_{1}}{z_{4}}\\ z_{1}=z_{4}=z_{cl}=f \end{array}\right.$$
where the coordinate of *F_cl_* is (0, 0, *ƒ*). After calculating the coordinates of point *F*_4_, then it is obtained from the point *F*_4_ and *F_h_* are co-linear:(17)$$\left\{\begin{array}{c} \frac{x_{4}}{x_{h}}=\frac{y_{4}}{y_{h}}=\frac{z_{4}}{z_{h}}\\ z_{4}=f\\ z_{h}=h \end{array}\right.$$

From Equations (16) and (17), the coordinates of points *F*_4_ and *F_h_* can be obtained in the same way. In the right virtual camera coordinate system, the coordinate of the right camera optical center on the left virtual camera coordinate system is *O_r_*(*a*, 0, 0). After this, it can be obtained as *F*_9_(*x*_9_, *y*_9_, *z*_9_):(18)$$\left\{\begin{array}{c} \frac{x_{9}-a}{x_{h}-a}=\frac{y_{9}}{y_{h}}=\frac{z_{9}}{z_{h}}\\ z_{9}=f \end{array}\right.$$

The extension of the line *F_h_F*_5_ intersects the right virtual imaging surface at point *F*_8_. The points *F*_7_, *F*_8_, *F*_9_, and the center of distortion *F_cr_* are collinear, and the line of this collinearity is *L_r_*_1_. Therefore, it can be obtained that:(19)$$\left\{\begin{array}{c} \frac{x_{8}-x_{9}}{x_{cr}-x_{9}}=\frac{y_{8}-y_{9}}{y_{cr}-y_{9}}=\frac{z_{8}-z_{9}}{z_{cr}-z_{9}}\\ y_{8}=y_{4}\\ z_{8}=z_{4}=f \end{array}\right.$$

The solution for the coordinates of the distortion center *F_cr_* in Equation (19) can be referred to Section 3.1.1. After calculating the coordinates of *F*_8_(*x*_8_, *y*_8_, *z*_8_), the coordinates of *F*_7_(*x*_7_, *y*_7_, *z*_7_) can be obtained in the same way. Any point on the left virtual imaging plane can obtain the corresponding epipolar line on the right virtual imaging plane.

### 3.3. Triangulation Modeling of Binocular Vision System under Non-Parallel Plane Refraction

The traditional triangulation method based on a pinhole camera model is ineffective in binocular vision measurement due to the plane refraction effect. This section analyzes the binocular vision measurement system under independent refractive planes. The image points of any space point *P* on the left and right imaging planes are *P_l_* and *P_r_*, respectively. The refraction rays *O_l_P_l_* and *O_r_P_r_* are not necessarily in the same plane. This leads to the failure of the conventional triangulation method based on the pinhole camera model. The camera rotation model and the camera imaging model under non-parallel plane refraction are proposed in Section 3.1.2. The model corrects the refraction error under non-parallel plane refraction. On this basis, this paper establishes a binocular vision system triangulation model under independent refractive planes.

In Figure 7, when the two refractive surfaces are independent, the camera coordinate system should be first corrected with rotation. A triangulation model of a binocular vision system with an independent refraction plane is established based on the virtual camera coordinate system. The distortion centers of the left and right virtual imaging surfaces are *W_cl_* and *W_cr_*, respectively. The image points of any space point *W* on the left and right virtual imaging surfaces are points *W*_1_ and *W*_7_, respectively. The reverse extension of line *W*_3_*W* intersects with the left virtual imaging surface at point *W*_4_. The reverse extension of line *W*_5_*W* intersects with the right virtual imaging surface at point *W*_8_. Triangulation modeling of binocular vision systems with independent refraction planes is based on a virtual camera coordinate system. From the previous analysis, points *W*, *W*_4_, and *W*_8_ are coplanar.

Triangular measurement of a binocular vision system under independent refractive surfaces can be achieved by using ray coplanarity as a constraint. The coordinates of points *W*_4_ and *W*_8_ can be solved from Equation (16).

### 3.4. Three-Dimensional Measurement Solution Process under Plane Refraction

The main flow of this paper is shown in Figure 8. The refraction plane normal vector ***n*** is solved by capturing before and after images of the refraction at the same attitude of the calibration plate. The camera rotation model is established based on the Rodriguez formula. The camera coordinate system is rotated and corrected to obtain the virtual camera coordinate system. The coordinates of *Q*_1_ in the virtual space coordinate system can be obtained by combining the equations of the measurement plane and the line *L* in the virtual camera coordinate system. The coordinates in the camera coordinate system are obtained from Equation (15). Both the left and right imaging planes undergo rotation correction, and binocular measurement is accomplished by employing the epipolar constraint model and the triangulation model under refraction effects.

The corner coordinates of the checkerboard image under the refraction effect are known, and the corresponding coordinate points in the virtual coordinate system can be obtained through simultaneous Equations (12) and (13). In Figure 7, the *W*_1_ and *W*_7_ coordinate points under the virtual coordinate system can be obtained. Further, the point coordinates of *W*_4_ and *W*_8_ in the virtual coordinate system can be solved by Equation (16) in Section 3.2. Finally, binocular measurements are realized using the refraction plane imaging epipolar constraint modeling and the triangulation model.

## 4. Experimentation and Analysis

A series of physical experiments were conducted to validate the effectiveness of the proposed method. Section 4.1 presented the experimental setup. Section 4.2 conducted experimental verification of camera imaging models under non-parallel plane refraction. Section 4.3 studied the 3D measurement experiment of binocular vision under plane refraction. The camera calibration was performed using the most common Zhang’s calibration method [24,25].

### 4.1. Experimental Setups

To validate the method proposed in this paper, a set of experimental platforms for the vision scheme of the tool change robot was built [26]. This is shown in Figure 9. It consists of a binocular camera, plate glass, target, motion module, and robot. Details are as follows:

(1)The camera model is Cognex CIC-1300, which boasts a resolution of 1280 pixels × 1024 pixels. The focal length of the camera lens is 12.5 mm, and the field of view angle is 25°. The measurement volume is 0.08 m^3^. The camera baseline distance is 205 mm. The distance between the checkerboard calibration plate and the camera optical center is approximately 610 mm.(2)The factory-measured refractive index [27] of K9 optical glass is 1.5437, with a thickness of 20 mm, which is considered as the true value.(3)The number of target points is 5 × 6, with a distance of 39 mm between adjacent points.(4)The motion module in Figure 9 can move in the left and right directions. Before the experiment starts, the motion module can move the target near the working distance of 600 mm. The motion module of the platform does not require high motion accuracy.

### 4.2. Experimental Validation of Camera Imaging Model under Non-Parallel Plane Refraction

This section verifies the positioning accuracy and measurement accuracy of the camera imaging model (hereinafter referred to as the refractive imaging model) under non-parallel plane refraction. Each set of experiments kept the last target attitude fixed, and images without and with plate glass were collected separately. To compare the positioning error and measurement error under different distortion center positions, the experimental section conducted two sets of experiments with different glass postures: (a) Glass attitude 1 and (b) Glass attitude 2. Figure 10, Figure 11, Figure 12, Figure 13 and Figure 14 show experimental data under two different glass postures. 

Figure 10 shows the distribution of image deviations under non-parallel plane refraction. Arrows indicate the direction of distortion. As shown in Figure 10a, the minimum deviation of the image points is 0.5 pixels, and the maximum deviation is 9.2 pixels. As shown in Figure 10b, the minimum deviation of the image points is 2.3 pixels, and the maximum deviation is 12.1 pixels. From the figure, it is evident that the distortion center is not the main point of the camera. The distortion center is shifted because the camera’s optical axis is not parallel to the normal vector of the refraction plane of the plate glass.

In Figure 11, the cyan point is the distortion center point. The distortion centers of the two experimental groups obtained by the least squares method are (953.07, 432.1) and (92.2, 395.81), respectively. In the two experiments shown in Figure 11, due to the difference in glass placement attitude, there was a significant change in the intersection point between the normal vector passing through the optical center and the image plane. The normal vectors of the refraction plane for the two sets of experiments are ***n*_1_** = (0.1268, −0.0284, 1) and ***n*_2_** = (−0.2262, −0.0431, 1). The camera rotation model corrects for the non-parallelism of the camera optical axis and the normal vector of the refraction plane. Figure 12 shows the image points before and after rotational correction with glass refraction.

To verify the localization accuracy of the refraction imaging model, both the pinhole camera model and the refraction imaging model are utilized to measure and compare the points on the 2D measurement plane. As shown in Figure 13, *P*1 represents the pinhole camera model under glass attitude 1. *R*1 represents the refraction imaging model under glass attitude 1. *P*2 represents the pinhole camera model under glass attitude 2. *R*2 represents the refraction imaging model under glass attitude 2. The two experiments are shown in Figure 13. The maximum localization errors of the pinhole camera model are 2.2707 mm and 3.1139 mm, respectively. The maximum localization errors of the refracted imaging model are 0.2428 mm and 0.3394 mm, respectively. The RMSE of positioning errors using the pinhole camera model are 1.3441 mm and 1.9937 mm. The RMSE of localization errors using the refraction imaging model are 0.1135 mm and 0.1484 mm, respectively.

Then, the measurement accuracy of the refraction imaging model is verified. To comprehensively reflect the impact of plane refraction on measurement accuracy, pinhole camera models and refractive imaging models are used to measure line segments *L*1, *L*2, *L*3, *R*1, *R*2, and *R*3 on the two-dimensional measurement plane. The measured line segments are shown in Figure 14.

In Table 1, the length of the line segment measured without glass is taken as the true value. The measurements of the pinhole camera model and the refraction imaging model are compared with the true value. The measurement results of two sets of experiments are shown in Table 1 and Table 2. The RMSE of the measurement errors using the pinhole camera model are 2.3821 mm and 2.4037 mm. The RMSE of the measurement errors using the refraction imaging model are 0.2130 mm and 0.2136 mm, respectively. From the above analysis, it can be concluded that the refraction imaging model can significantly improve measurement accuracy.

In Table 2, the actual dimensions between the target points of the marker are taken as the real values. It can evaluate the actual measurement accuracy of the proposed method. The RMSE of the measurement errors using the pinhole camera model were 2.2827 mm and 2.4179 mm, while the RMSE of the measurement errors using the refraction imaging model were 0.3929 mm and 0.2627 mm, respectively. From the above analysis, it can be concluded that the refraction imaging model can significantly improve measurement accuracy.

### 4.3. Experiments on 3D Measurement of Binocular Vision under Plane Refraction

First, the Zhang calibration method calibrates the internal and external parameters of the two cameras. The method proposed solves the refraction plane normal vector. Table 3 demonstrates the calibration parameters. Where (*f_x_*, *f_y_*) is the camera focal length, (*u*_0_, *v*_0_) is the coordinates of the camera main point, *kc* is the distortion coefficient, ***M_R_*** is the rotation matrix, ***M_T_*** is the translation matrix, (*u_c_*, *v_c_*) is the distortion center, and ***n*** is the normal vector of the refraction plane of the plane glass. Find the center of distortion and the normal vector of the refraction plane. In Figure 15, the cyan point is the center distortion point for the left and right cameras.

#### 4.3.1. Experimental Verification of Epipolar Constraint Model for Independent Refractive Plane Imaging

After calibration of the internal and external parameters of the binocular vision system is completed, the independent refraction plane imaging epipolar constraint model is experimentally validated. It has the same basic principles as the shared refraction plane imaging epipolar constraint model and has a wider range of applications. Therefore, this paper only analyzes the epipolar constraint model of independent refractive plane imaging.

In the experiment, keeping the optical axis of the left and right cameras not parallel to the normal vector of the refractive plane, the cameras collect one image with the plate glass and one image without the plate glass.

To utilize a binocular vision system for 3D measurement, it is essential to initially conduct feature point matching between the left and right images. However, Figure 16 demonstrates that the polar constraint principle of traditional binocular vision systems is rendered ineffective due to the influence of the plane refraction effect. The green line represents the limit constraint. In Figure 17, Position 1 indicates a large difference in the longitudinal tilt angle of the left and right plate glass. In Figure 17, Position 2 indicates that the left and right plate glass are parallel.

In Figure 17, when the tilt angle difference between the left and right plate glass in the longitudinal direction is large, the matching error without refraction is less than 1 pixel, and its RMSE is 0.2343 pixels. The RMSE of the matching error in the presence of refraction is 11.1107 pixels. The traditional principle of polar constraints is no longer applicable to the binocular vision systems affected by plane refraction.

But it’s not that the polar constraint principle will fail as long as there is refraction. When the left and right plate glass are nearly parallel in the longitudinal direction, the refraction has minimal impact on the coordinates *v* of the image points. The normal vector of the refractive surface of the left plate glass is ***n_l_*** = (−0.1181, −0.2283, 1). The normal vector of the refractive surface of the right plate glass is ***n_r_*** = (0.0063, −0.2396, 1). When the left and right plate glass are nearly parallel in the longitudinal direction, The matching error with refraction is less than 1 pixel, and the RMSE of the matching error with refraction is 0.7573 pixels. The traditional polar constraint principle remains applicable in this situation. In Figure 17, Position 1 indicates a large difference in the longitudinal tilt angle of the left and right plate glass. In Figure 17, Position 2 indicates that the left and right plate glass are parallel.

#### 4.3.2. Experimental Validation of a Triangulation Model for Binocular Vision System under Independent Refraction Planes

Finally, the proposed triangulation model of the binocular vision system under an independent refraction plane is experimentally validated. It compares depth-of-field errors, positioning errors, and measurement errors for space points with and without considering refraction effects. Figure 18a shows the comparison of depth-of-field between the checkerboard corners before and after correction, where the depth-of-field calculated without refraction is taken as the true value. Figure 18b shows the comparison of the depth-of-field error before and after correction. The RMSE of the depth-of-field error before and after correction are 2.9902 mm and 0.3187 mm, respectively.

Figure 18b shows the comparison of positioning errors of checkerboard corner points before and after correction. The 3D coordinates calculated without refraction are used as the true spatial point coordinates. Figure 19 shows that the RMSE of the positioning errors before and after correction are 3.5661 mm and 0.3465 mm, respectively.

## 5. Discussion

This paper presents a complete vision measurement scheme applicable to the presence of plate glass refraction effects. The effectiveness of the scheme was verified through a series of experiments.

### 5.1. Analysis of the Proposed Method

In Section 3.2, the normal vector n of the refractive surface is used to describe the camera imaging model accurately when the camera’s optical axis is not parallel to the normal vector of the refractive plane. The distortion center is used to obtain the normal vector of the refraction plane of the plate glass based on the expansion offset effect of plane refraction. The established model for image distortion correction under plane refraction realizes the correction of image distortion. This model can not only eliminate image distortion but also intuitively reflect the impact of different incident angles and depth of field on measurement error.

### 5.2. Stability Experiment under Simulated Tunnel Construction Environments

Two types of stains shown in Figure 19 were attached to the outer surface of the plate glass to analyze how the tunnel construction environment affects the proposed plate glass calibration method. Refer to previous experiments on glass pollution. The first experiment (Non-pollution) is conducted under the ideal condition where glass is not polluted. The second experiment (Pollution A) simulated the condition of glass polluted by sewage. The third experiment (Pollution B) simulates the condition where glass is polluted by sludge and partially obstructed. According to the above experimental scheme, the calibration experiment of the refraction plane of plate glass under the simulated tunnel construction environment is carried out.

The plate glass calibration is accomplished using target points that are unpolluted or only slightly polluted. The experimental results are shown in Table 4.

According to Table 4, compared to the Non-pollution condition, the cosine similarity of the unit normal vector of the refractive surface under both polluted conditions is close to 1. This shows that pollution A and pollution B have less of an effect on the solution accuracy of the proposed method.

### 5.3. Future Work

Section 4.3 conducted stability experiments in simulated tunnel construction environments. However, the real tunnel construction scenes are even more harsh. Image restoration using deep learning techniques serves as a set of potential solutions. The plate glass used for the experiments in this paper is *K*9 optical glass. However, in practical applications, some explosion-proof tempered glass will be used, whose glass performance may not be as stable as optical glass. In the future, its practical application scenarios can be considered, and several experiments can be conducted using glass with different parameters to verify the stability of the model.

## 6. Conclusions

To overcome the failure of conventional vision measurement techniques under plate glass refraction, this paper provides an effective and stable solution for refraction imaging modelling, glass parameter calibration, and 3D coordinate solutions. It analytically realizes the correction of refractive distortion images. The glass normal vector is obtained from the calibration. When the camera’s optical axis is not parallel to the normal vector of the refractive plane, a camera rotation model and a camera imaging model under non-parallel plane refraction are established to correct refractive errors. In particular, an independent refractive plane triangulation model is proposed to address the issue of triangulation failure under planar refraction. This experiment verifies the positioning accuracy and measurement accuracy of the camera imaging model under non-parallel plane refraction. These experimental results show that the RMSE of the positioning errors before correction are 1.3441 mm and 1.9937 mm. The positioning errors after correction are 0.1135 mm and 0.1484 mm, respectively. The measurement errors before correction are 2.3821 mm and 2.4037 mm. The measurement errors after correction are 0.2130 mm and 0.2136 mm, respectively. This experiment verifies the accuracy of the binocular vision system triangulation model under independent refractive planes. The RMSE of the depth of field error before and after correction are 2.9902 mm and 0.3187 mm, respectively. The RMSE of the positioning errors before and after correction are 3.5661 mm and 0.3465 mm, respectively. From the above analysis, the proposed triangulation model under independent refractive planes greatly improves the accuracy of depth-of-field and positioning. This proves the effectiveness of the proposed model. While this solution is designed for tool changer robot vision systems in harsh construction environments, it can serve as a reference for any vision measurement application under glass window protection.

## Figures and Tables

**Figure 1 sensors-24-00066-f001:**
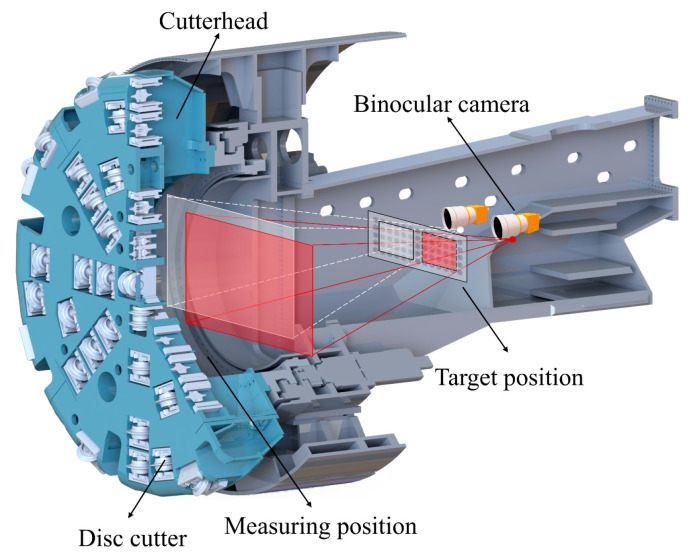
The TBM diagram of the changing area.

**Figure 2 sensors-24-00066-f002:**
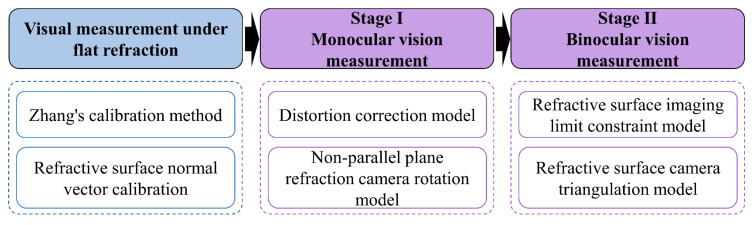
The basic content of this article.

**Figure 3 sensors-24-00066-f003:**
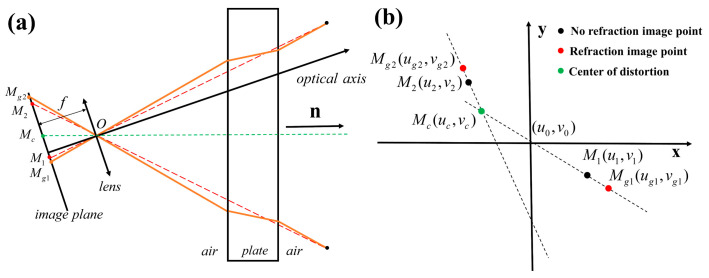
An Imaging model when the camera’s optical axis is not parallel to the normal vector of the refraction plane. (**a**) Plane refraction imaging and (**b**) image points on the imaging surface.

**Figure 4 sensors-24-00066-f004:**
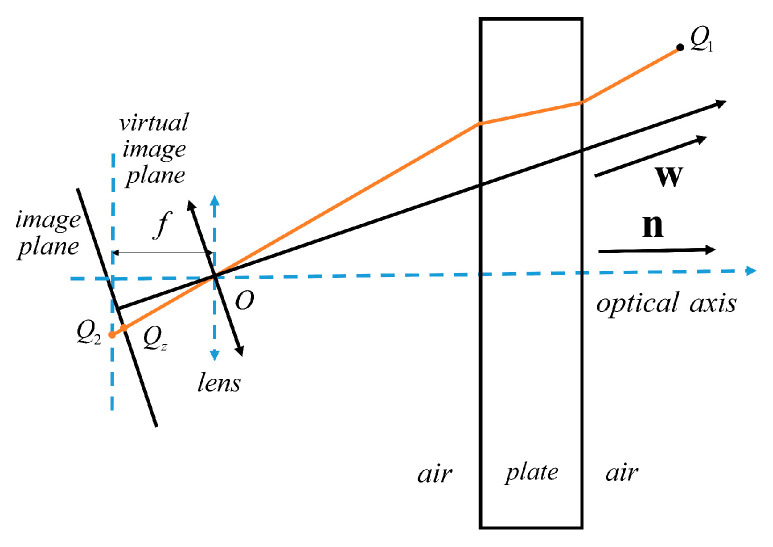
Non-parallel plane refractive camera rotation model.

**Figure 5 sensors-24-00066-f005:**
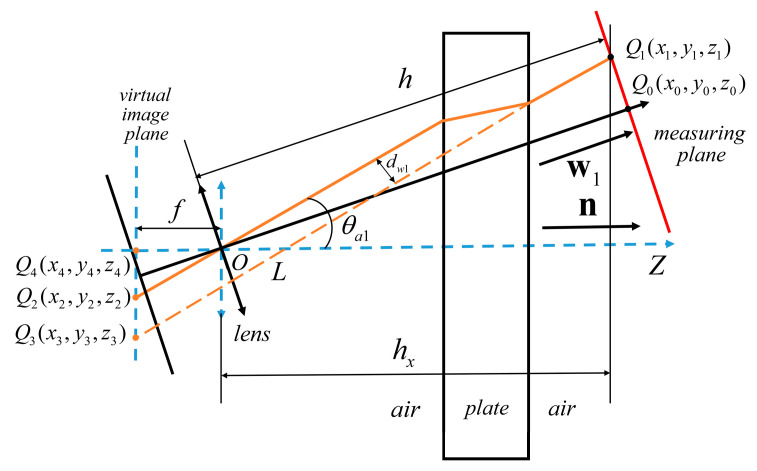
Non-parallel plane refractive camera imaging model.

**Figure 6 sensors-24-00066-f006:**
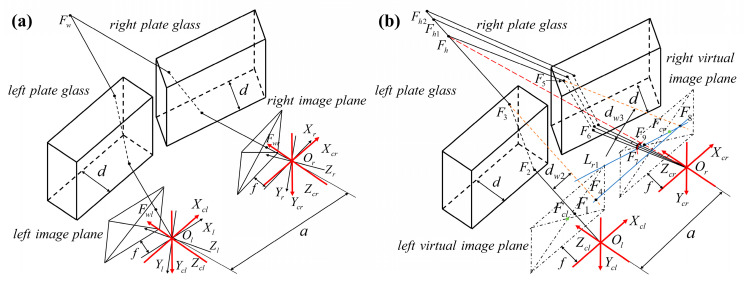
Independent refractive surface (**a**) imaging model and (**b**) epipolar constraint model.

**Figure 7 sensors-24-00066-f007:**
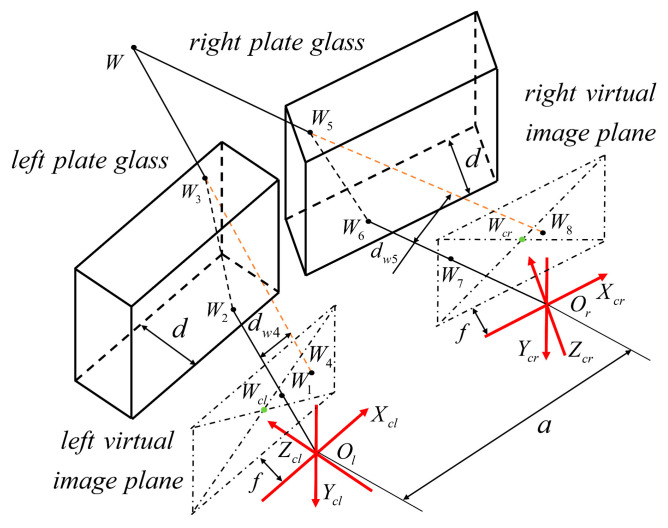
Triangulation measurement model under an independent refraction plane.

**Figure 8 sensors-24-00066-f008:**
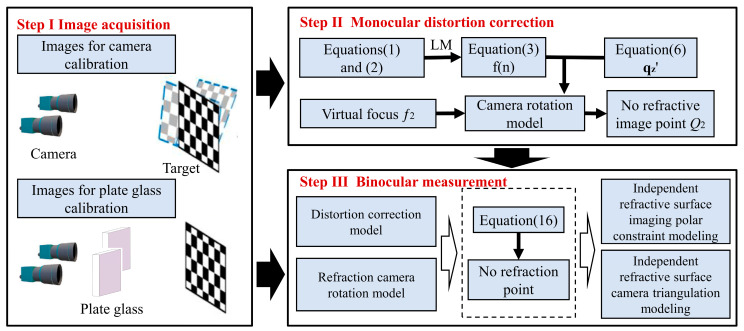
Overall flow chart of binocular measurement under plate refraction.

**Figure 9 sensors-24-00066-f009:**
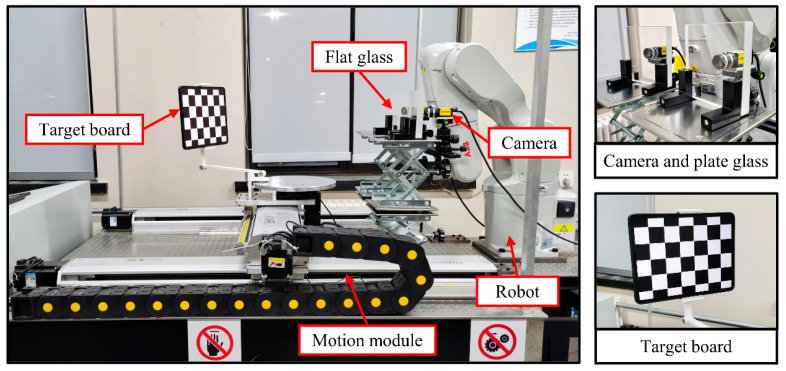
Experimental platform for vision program of the tool-change robot.

**Figure 10 sensors-24-00066-f010:**
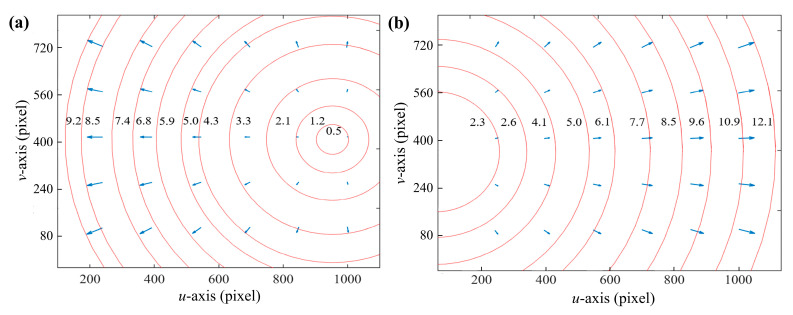
Image deviation distribution plot. (**a**) Glass attitude 1 and (**b**) glass attitude 2.

**Figure 11 sensors-24-00066-f011:**
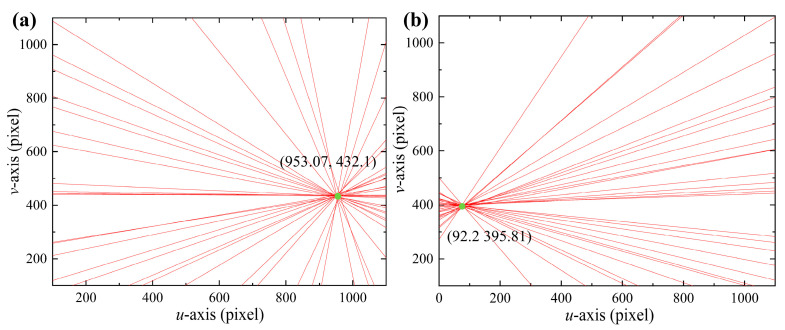
Determination of the distortion center. (**a**) Glass attitude 1 and (**b**) glass attitude 2.

**Figure 12 sensors-24-00066-f012:**
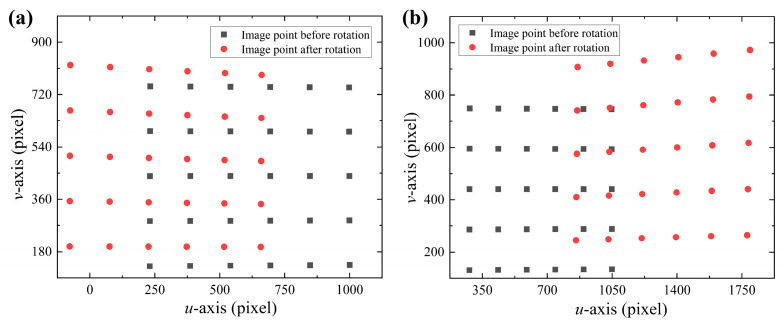
Image point comparison before and after camera rotation with glass. (**a**) Glass attitude 1 and (**b**) glass attitude 2.

**Figure 13 sensors-24-00066-f013:**
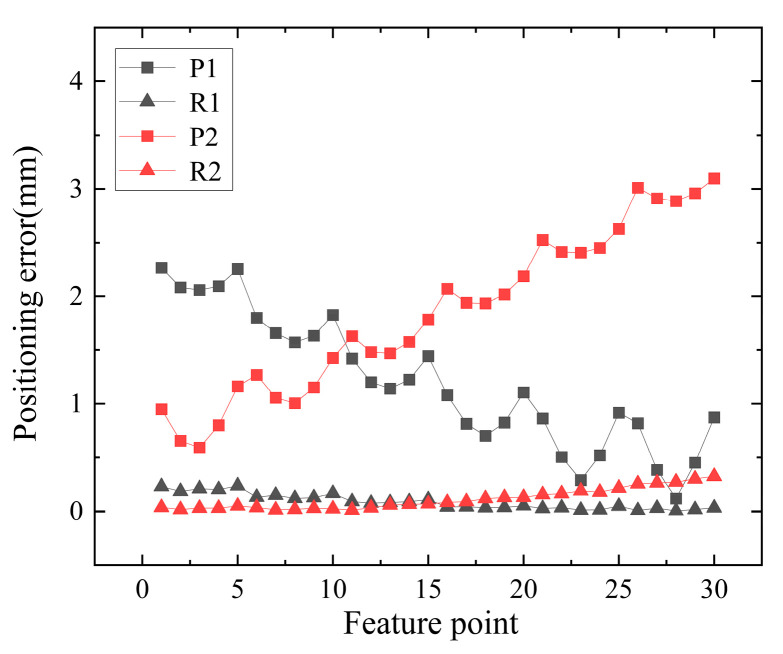
Pinhole camera model versus refractive imaging model positioning error.

**Figure 14 sensors-24-00066-f014:**
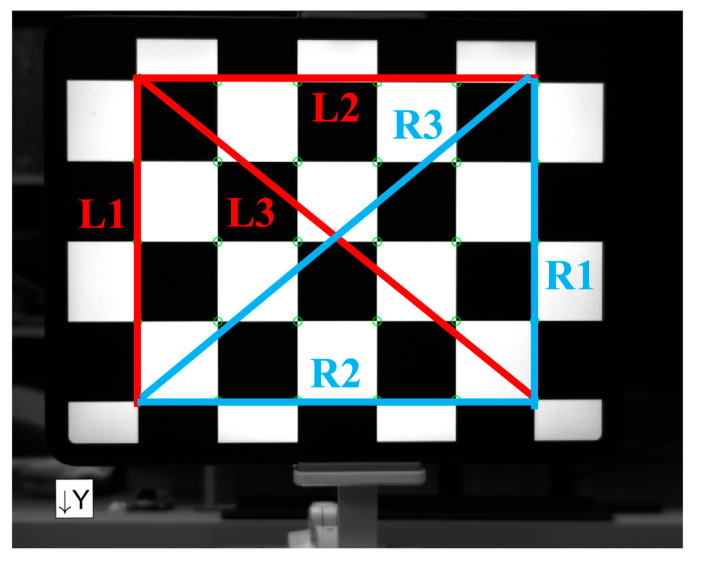
The measured segment position.

**Figure 15 sensors-24-00066-f015:**
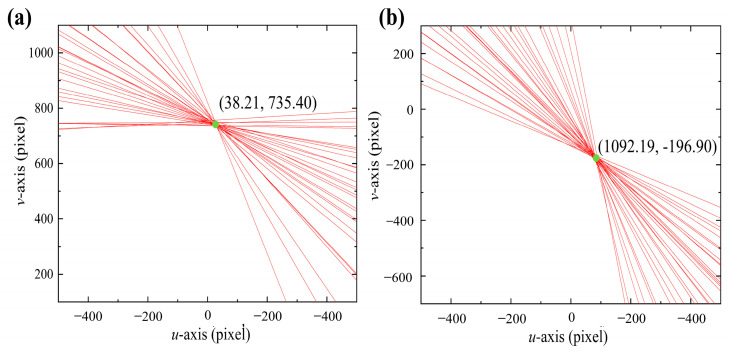
Determination of the distortion center. (**a**) Left camera and (**b**) right camera.

**Figure 16 sensors-24-00066-f016:**
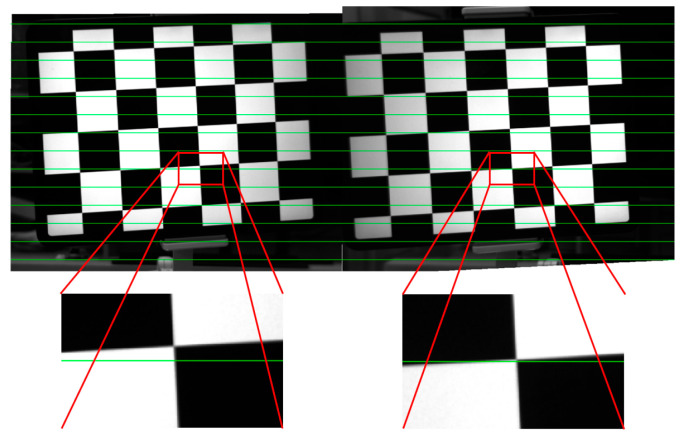
Epipolar correction diagram with glass.

**Figure 17 sensors-24-00066-f017:**
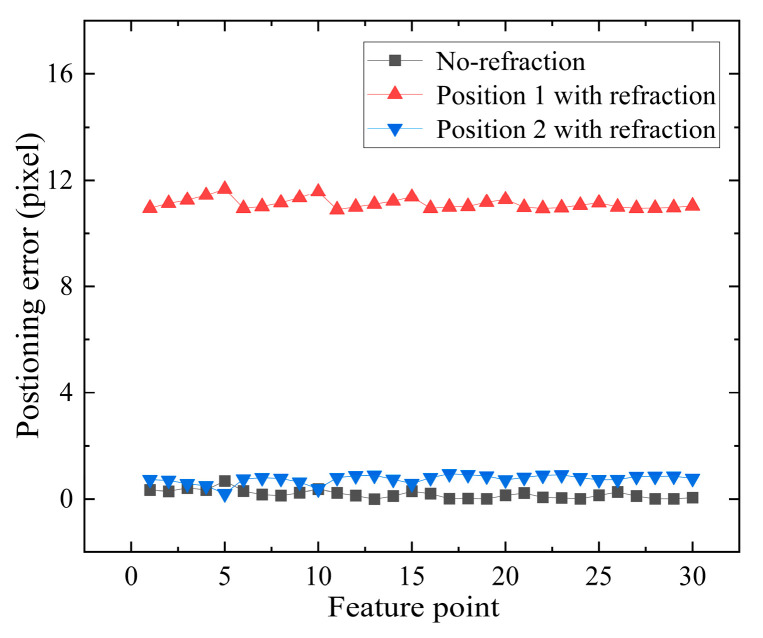
Matching error with and without refraction at different glass positions.

**Figure 18 sensors-24-00066-f018:**
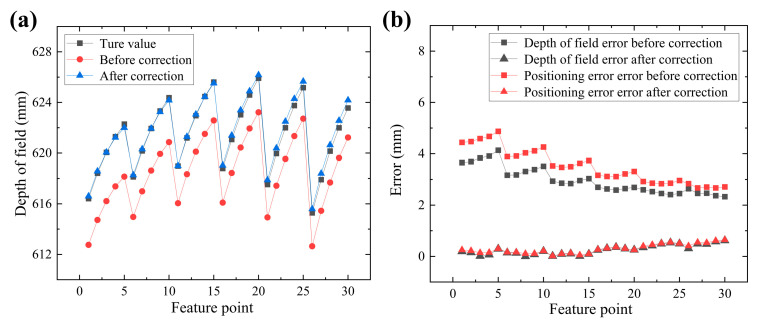
Comparison before and after calibration. (**a**) Depth of field and (**b**) depth of field error and positioning error.

**Figure 19 sensors-24-00066-f019:**
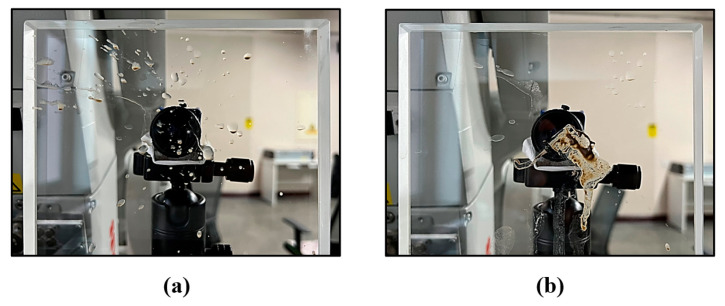
Pollution type diagram. (**a**) Glass is polluted by sewage. (**b**) Glass is polluted by sludge.

**Table 1 sensors-24-00066-t001:** Measurement error in Experiment 1. Units: mm.

Group	Line Segment	Real Value	Pinhole Camera Model	Refraction Imaging Model	Pinhole Camera Model Measurement Error	Refraction Imaging Model Measurement Errors
experiment one	L1	156.6668	158.5312	156.8691	1.8645	0.2023
	L2	194.8575	197.1360	195.0896	2.2785	0.2321
	L3	249.9980	252.8859	250.2325	2.8978	0.2445
	R1	155.6221	157.3648	155.6632	1.7427	0.0411
	R2	194.9240	197.2314	195.1439	2.3074	0.2199
	R3	249.0547	251.9907	249.3145	2.9360	0.2598
experiment two	L1	156.5188	158.2746	156.4472	1.7558	0.0716
	L2	194.7974	197.1148	195.0683	2.3175	0.2710
	L3	249.9564	252.8827	250.1399	2.9263	0.1835
	R1	155.7484	157.6053	155.9255	1.8569	0.1770
	R2	194.8289	197.1602	195.0612	2.3313	0.2323
	R3	249.9526	252.9136	250.2288	2.9610	0.2762

**Table 2 sensors-24-00066-t002:** Measurement error in Experiment 2. Units: mm.

Group	Line Segment	Real Value	Pinhole Camera Model	Refraction Imaging Model	Pinhole Camera Model Measurement Error	Refraction Imaging Model Measurement Errors
experiment one	L1	156.0000	158.5312	156.8691	2.5312	0.8691
	L2	195.0000	197.1360	195.0896	2.1360	0.0896
	L3	249.7218	252.8859	250.2325	3.1641	0.5107
	R1	156.0000	157.3648	155.6632	1.3648	0.3368
	R2	195.0000	197.2314	195.1439	2.2314	0.1439
	R3	249.7218	251.9907	249.3145	2.2689	0.4073
experiment two	L1	156.0000	158.2746	156.4472	2.2746	0.4472
	L2	195.0000	197.1148	195.0683	2.1148	0.0683
	L3	249.7218	252.8827	250.1399	3.1609	0.4181
	R1	156.0000	157.6053	155.9255	1.6053	0.0745
	R2	195.0000	197.1602	195.0612	2.1602	0.0612
	R3	249.7218	252.9136	250.2288	3.1918	0.5070

**Table 3 sensors-24-00066-t003:** Camera parameters.

	Left Camera	Right Camera
(f_x_, f_y_)/mm	(2438.07, 2438.08)	(2443.13, 2443.13)
(u_0_, v_0_)/pixel	(643.98, 500.01)	(647.83, 487.49)
kc	(−0.4082, 0.4359)	(−0.4100, 0.4389)
M_R_	$\left[\begin{array}{c} \begin{array}{ccc} 0.9572 & 0.0008 & -0.2893 \end{array}\\ \begin{array}{ccc} -0.0087 & 0.9996 & -0.0262 \end{array}\\ \begin{array}{ccc} 0.2892 & 0.0275 & 0.9569 \end{array} \end{array}\right]$	
M_T_	[−211.4756, 1.3306, 53.3775]^T^	
(u_c_, v_c_)/pixel	(38.21, 735.40)	(1092.19, −196.90)
n	(−0.2483, 0.0961, 1)	(0.1823, −0.2809, 1)

**Table 4 sensors-24-00066-t004:** Unit normal vectors and refractive indices.

Serial Number		Unit Normal Vector	Cosine Similarity
Left glass	Non-pollution	[−0.0500, −0.0475, 0.9976]^T^	/
	Pollution A	[−0.0478, −0.0466, 0.9973]^T^	0.9999972
	Pollution B	[−0.0475, −0.0463, 0.9977]^T^	0.9999961
Right glass	Non-pollution	[−0.0036, −0.0235, 1.0000]^T^	/
	Pollution A	[−0.0040, −0.0238, 0.9998]^T^	0.9999999
	Pollution B	[−0.0043, −0.0212, 1.0006]^t^	0.9999972

## Data Availability

The authors confirm that the data supporting the findings of this study are available from the corresponding author, upon reasonable request.

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
