# Peer review of "Vision Measurement Method Based on Plate Glass Window Refraction Model in Tunnel Construction"

_sensors, 2023, doi:10.3390/s24010066_

Round 1

Reviewer 1 Report

Comments and Suggestions for Authors

This paper proposes the image distortion correction model, the refraction plane imaging epipolar constraint model, and the refraction plane triangulation model. Besides, a series of physical experiments is conducted to verify the validity of those models.

A complete and exhaustive review from a native speaker is strongly suggested.

Some grammar corrections are urgently needed. (such as the sentences described in Line 257-259, 261-262, 323-324, and so on)

Several sentences need to be more concise. (such as the sentences described in Line 82-85, 156-158, and so on)

The abbreviation (TBM, Line 28) should be described its meaning, firstly.

Where is Sec. 6 described Line 92?

The vector should be bold and italic. (such as ‘n’ described in Line 107, ‘q1’ described in Line 198, and so on)

What’s the special meaning of the single sentence “In practical implementation.” described in Line 117?

Where are the ‘Q4’ point and ‘w1’ vector shown in Fig. 5?

Where is the ‘O1F1’ line shown in Fig. 6?

Some latter letters or numbers should be subscript, such as ‘Q4’ in Line 163, ‘F4’ in Line 221, ‘Fh’ in Line 231, and so on.

Is there any reference to prove that the refractive index of the K9 glass is 1.5437? As far as I know, the refractive index of glass is about 1.5163.

Why is exactly the thickness of experimental glass 20mm? Is there any reference to prove that above glass is enough to withstand the high pressure?

What is the motion accuracy of the Motion module described in Fig. 9? How to offset the error caused by the Motion module?

Authors need to describe the experimental platform on more detail. What is the measurement volume, baseline, working distance, FOV, and so on?

What are the differences between Figs. 10-14(a) and Figs. 10-14(b) respectively? It is not described in the captions of Figs. 10-14.

What is the reason regarding the huge difference about distortion center points between Fig. 11(a) and Fig. 11(b)?

What is the difference between Table 1 and Table 2? It is not described in the caption.

What is the difference between Position 1 and Position 2 described in Fig. 17?

How to calculate and obtain the depth of field before and after correction?

Is the measurement system feasible in the actual TBM environment?

The true value of above experiment is calculated when there is no refraction. Is it feasible and schematic?

Reviewer 2 Report

Comments and Suggestions for Authors

It is necessary to consider the influence of protective lenses on the amount of light entering the text, and at the same time standardize the icons in the text, especially the amount of information in the result diagram is too small, and similar diagrams can be merged.

Comments on the Quality of English Language

English expressions need to be simplified

Reviewer 3 Report

Comments and Suggestions for Authors

This paper proposes a correction model for plane refraction. Their experimental results validate the correctness of the model and show significant reduction of measuring errors. Basically, the paper is technically sound. But couple points should be clarified.

-          In line 86, “Section 2 introduces the related work.”. But no “related work section” is found.

-          The related work provided in the introduction section (lines 50-70) is too brief to give enough background understanding of the challenge of the problem. Therefore, it is difficult for readers to easily justify the contribution of this work.

-          Line 117 is a dangling sentence.

-          In Line 141, vector “v” should be in boldface.

-          In in Figure 5, Q4 is missing.

In Line 174, “w1” should be in boldface

Comments on the Quality of English Language

None

Round 2

Reviewer 1 Report

Comments and Suggestions for Authors

The author made a careful revision and gave a detailed response, and I think this manuscript is ready for acceptance.